# Cybersecurity in the Internet of Things in Industrial Management

**Ricardo Jorge Raimundo** [1,2,*] **and Albérico Travassos Rosário** [3]

1    ISEC Lisboa, Instituto Superior de Educação e Ciências, 1750-142 Lisboa, Portugal
2    IPAMLab, Estrada da Correia, 53, 1500-210 Lisboa, Portugal
3    GOVCOPP-Governance, Competitiveness and Public Policies, IADE—Faculdade de Design,
     Tecnologia e Comunicação, Universidade Europeia, 1500-210 Lisboa, Portugal; alberico@ua.pt
*    Correspondence: ricardo.raimundo@iseclisboa.pt

**Abstract:** Nowadays, people live amidst the smart home domain, while there are business opportunities in industrial smart cities and healthcare. However, there are concerns about security. Security is central for IoT systems to protect sensitive data and infrastructure, whilst security issues have become increasingly expensive, in particular in Industrial Internet of Things (IIoT) domains. Nonetheless, there are some key challenges for dealing with those security issues in IoT domains: Applications operate in distributed environments such as Blockchain, varied smart objects are used, and sensors are limited, as far as machine resources are concerned. In this way, traditional security does not fit in IoT systems. The issue of cybersecurity has become paramount to the Internet of Things (IoT) and the Industrial Internet of Things (IIoT) in mitigating cybersecurity risk for organizations and end users. New cybersecurity technologies/applications present improvements for IoT security management. Nevertheless, there is a gap in the effectiveness of IoT cyber risk solutions. This review article discusses the literature trends around opportunities and threats in cybersecurity for IIoT, by reviewing 70 key articles discovered from a profound Scopus literature survey. It aims to present the current debate around the issue of IIoT rather than suggesting any particular technical solutions to solve network security problems.

**Keywords:** Internet of Things; cybersecurity; industry 4.0; security data; machine learning; blockchain; cloud computing; network computing

## 1. Introduction

The Internet of Things (IoT) aims to integrate the digital and physical universes into a distinct system, providing major business opportunities for several sectors such as industry, tourism, and energy. It has created a new paradigm in which a network of machines and devices capable of communicating and collaborating with each other are driving new processes. However, the IoT is fragile regarding many security issues that are often highly demanding due to its complex context and a vast number of tools, which present flaws in terms of resources [1]. The IoT is a system where the Internet is linked to the physical world through sensors [2]. It can be deemed as the management of a network of devices, home appliances, and vehicles of the IoT, and is challenging due to the dynamic nature of the linkage between devices, actors, and resource constraints [3,4] involving hardware, software, sensors, and connectivity that allows them to connect, gather, and exchange data [5]. At the core of the IoT is the "smart factory", which comprehends several different elements: person, process, intelligent object, and technological ecosystem [6]. The IoT embraces traditional internet connectivity to likewise traditionally non-physical devices such as cars and electric tools, to name but a few. The IoT is also strongly related to manufacturing, in order to produce high-quality products at low cost by assembling the Industrial Internet of Things (IIoT), cloud computing, and big data analytics [7], including robots [8].

The IoT has therefore become prevalent over diverse sectors, with the Internet of Battlefield Things (IoBT) or the Internet of Vehicles (IoV) [3,9], along with its security issues, contributing to an increasing number of cyber-attacks. Consequently, there has been recent concern with cybersecurity in this domain, amid a lack of policy direction and a lack of understanding of user values related to cybersecurity in terms of the IoT, while policy has not been guided by key stakeholder values [10].

Cybersecurity is concerned with the protection of electronics, software, and data, along with the procedures by which systems are accessed. In general, security objectives comprise privacy, in terms of information not inappropriately disclosed to unauthorized devices or individuals to be modified or destroyed [11,12]. Hence, due to the countless existing IoT-based connected devices, society is also becoming increasingly vulnerable to cyber-attacks such as denial-of-service attacks [13] by hackers and insiders, for instance, that deny direct access to devices, etc. [14,15]. Technology is increasingly more central in our daily lives, which means that cybercrime and cybersecurity tools evolve concomitantly [16,17], across the whole manufacturing sector [18], which needs to invest in cybersecurity countermeasures, while new technologies are emerging for IoT cybersecurity management [19].

Furthermore, cyber-attacks on smart grids, as key infrastructure components are particularly vulnerable and bear greater costs, impacts severely on the safety of citizens and governments [13]. There is a growing concern towards cybersecurity and the lack of effective countermeasures, e.g., cybersecurity professionals [20]. China, for example, is creating new cybersecurity law and strategy [21]. Furthermore, while healthcare is currently a hot topic because there is an abundance of critical data, cyber defenses are on average weak in hospitals, where patients' lives and trust are at risk [22].

As previous literature focuses on the technical features of IoT cybersecurity, there is a knowledge gap on frameworks to address the complex cybersecurity issues in the IoT. This research provides a literature review on IoT security technologies and cyber risk management in industry.

The article is organized as follows. In Section 2, we put forward diverse theoretical concepts related to cybersecurity in the IoT. In Section 3, we present the methodological approach. In Section 4, we discuss the main fields of use of cybersecurity with regard to the IoT which have arisen from the literature. Finally, we conclude by suggesting implications and future research avenues.

## 2. Literature Review: Key Concepts

Due to the characteristics of both cyber-attacks and IoT systems, it is necessary to understand the discussed concepts before progressing to the major current trends on the issue.

### 2.1. Cybersecurity

Cybersecurity has become a major concern, as we know that many of our everyday objects can be connected to the Internet, which is paramount in our daily lives. If it can be connected, it can be accessed. Thus, the primary concern for cybersecurity relies upon intrusion detection [23], in which physical or cloud computer activities are monitored through analysis of system vulnerabilities and activity patterns [24]. Attacks may take the form of distributed denial of service (DDoS) [13], malicious IPs [25], and data manipulation [26], for instance, with ensuing outcomes such as loss of information, operational losses, and health damage [22,27].

### 2.2. The Internet of Things

As previously mentioned, the Internet of Things (IoT) can be described as a new theme that encapsulates both the prevailing internet and physical artifacts [1]. We can mention "smart homes" [11], for instance, referring to home automation, manufacturing systems as the industrial process, and health in terms of hospital automation [11].

In this vein, the IoT heavily augments the multiple gadgets and connected devices in our lives, for instance, in smart grids [11] and in transportation through electric vehicles (EV) [3]. Thus, the internet technology, although presenting countless advantages, also poses serious threats [28]. IoT applications consequently cover a wide range of artifacts, from smart homes [25] to huge smart factories [6] to smart grids [13]. In each case, the correspondent devices are complemented with wireless interfaces of wireless sensor networks (WSN) that constitute a key IoT technology [1,2] to the wide stream of IoT systems. Examples include "smart grid", "Internet of Things", "manufacturing systems", "smart cities", and "cloud computing in transport and smart homes" [6,8,11,25].

On the one hand, in the case of smart homes, it is advisable to protect sensors' identities from being recognized through wireless communication environment networks, while keeping the software up to date from trustable vendors and cloud providers [1]. On the other hand, in the case of smart cities, to which many populations will tend to migrate, the IoT offers multiple services such as smart parking, environmental, waste, water and traffic management, and energy consumption monitoring, through operations encompassing the IoT spectrum, its energy and architecture efficiency, and mitigating its environmental effects, keeping in mind its context interplay [26,29].

## 2.3. The Industrial Internet of Things (IIoT)

The IIoT presents diverse nuances that differentiate it from the traditional IoT. While the IoT operates in domestic environs, the IIot operates in industrial environs. In this way, it involves the optimization of supply chains, for instance. The IIoT equals Industry 4.0 [30], which is a shared term for technologies and theories of value chain organization [18,31]. Industry 4.0 presents a modular structure, through which computers monitor and manage smart factories and ensuing physical processes [32], creating a digital copy of the physical processes while making decentralized decisions [33]. Along the way, computer systems interact both with one another and with people [30].

Also, both organizational and interorganizational services can be provided to actors of the supply chain. Interconnected objects, managed and accessed through data mining processes such as Blockchain, can be partly accessed and function as sensors and are enabled to interact with other devices [34,35]. Such systems, made up of smart artifacts within the IoT system, demand minimal or no human action in order to exchange and produce data, often assisted by artificial intelligence mechanisms [36]. In summary, the IIoT's major concerns include reducing material and energy consumption, better managing the temporal dimensions of security in terms of "intrusion detection", cloud computing, and the interface between supply chain management and marketing processes, plus better managing the complexity of infrastructures in terms of the number of entry points [11,18,32,34,37,38].

The IIoT comprises both cybersecurity and IoT concerns in general. It focuses on integrity, in which data is protected from modification by unauthorized parties; authentication, in which the data source is verified as the pretended identity [39]; privacy, in which users' identities are non-traceable from their behaviors; [40] confidentiality, in which information is made unintelligible to unauthorized entities; and availability, in which the system services are available only for legitimate users [41].

IIoT thus faces important challenges, namely regarding operations in decentralized environs such as Blockchain systems [42,43] and the varying nature of smart artifacts [44]. In addition, it is worth mentioning the sparse computational resources and power available to the diverse sensors that result in insufficient traditional security measures [9,45]. The aforementioned issues increase the chances of cyber-attacks on IoT systems, namely plants, transport, and household appliances [9], demanding substantial improvement in terms of authentication from remote systems, encryption from new sensors, and web interface and computer software for intrusion detection [46]. Additionally, the more IoT innovation, the more developed wireless technologies are, as in the case of 5G, which is optimized well beyond voice and data, offering a vast array of opportunities [15,47].

The literature review presented here also suggests a set of security solutions for cordless sensor networks with respect to the IoT [48–50]. In particular, in terms of network computing, decentralized architectures made up of countless objects [15] such as Blockchain [25] and cloud computing systems ease network management and configuration [51,52], ameliorating IoT security [53] through sensors that optimize data sending, avoiding the redundancy in the wireless channels by systems that improve networking such as big data [18,30,54,55].

The design of the conceptual and technological framework for this article was not made randomly, but rather through a preliminary search on Scopus with the keywords "Internet of Things" and "Cyber Security", with the results shown and discussed in the following sections.

### 3. Materials and Methods

This investigation uses a Systematic Review of Bibliometric Literature (LRSB), as proposed by Rosário and Raimundo [56], Raimundo and Rosário [57], and Rosário et al., [58]. This qualitative approach analyzes and synthesizes documents on cybersecurity in the Internet of Things in Industrial Management that clearly indicates some contexts for the purpose of research through rigorous and precise design. It summarizes and combines relevant studies, thus expanding usable knowledge in decision-making strategies. The main advantage of qualitative research is to allow the collection and analysis of data on cybersecurity factors in the Internet of Things in Industrial Management. LRSB is designed to be methodical and explicit. This type of study provides guidance for the development of sketches, indicating new methods for future investigations, and identifies which research methods have been used. This methodology is designed to build new knowledge regarding the context of cybersecurity in the Internet of Things in Industrial Management.

The LRSB process was divided into three phases and six stages (Table 1), as proposed by Rosário and Raimundo (2021), Raimundo and Rosário (2021), and Rosário et al., (2021).

**Table 1.** The systematic LRSB process.

| Phase | Step | Description |
|---|---|---|
| Exploration | Step 1 | problem of research |
| | Step 2 | search of appropriate literature |
| | Step 3 | the critical precision of the chosen studies |
| | Step 4 | synthesis of data from individual sources |
| Interpretation | Step 5 | reports and recommendations |
| Communication | Step 6 | presentation of the LRSB report |

The database of indexation of scientific and/or academic documents was Scopus, the most important peer review of the scientific and/or academic environment. It lists nearly 19,500 titles from more than 5000 international publishers, covering 16,500 peer-reviewed journals in the scientific and/or academic fields.

However, we acknowledge that the study has the limitation of considering only the Scopus indexing database, excluding other scientific and academic indexing databases.

Bibliographic research includes peer-reviewed scientific and/or academic documents published until September 2021. The initial search involved the keywords "Cyber Security" and "Internet of Things" to track summaries, titles, and keywords. There were 15,748 documents identified that used the keyword "Cyber Security", and this was reduced to 1316 by adding the keyword "Internet of Things". The research was later limited to the research area "Business, Management, and Accounting" in order to obtain only the most relevant research (Table 2).

**Table 2.** The bibliographic research screening methodology.

| Database Scopus | Screening | Publications |
|---|---|---|
| Meta-search | keyword: Cyber Security | 15,748 |
| First Inclusion Criterion | keyword: Cyber Security, Internet of Things | 1316 |
| Second Inclusion Criterion | keyword: Cyber Security, Internet of Things subject area: Business, Management, and Accounting | 60 |
| Tracking | keyword: Cyber Security, Internet of Things subject area: Business, Management Published Until September 2021 | |

Source: own elaboration.

Finally, content techniques and thematic analysis were used to recognize, analyze, and report the various documents proposed by Rosário and Raimundo (2021), Raimundo and Rosário (2021), and Rosário et al., (2021).

The 60 scientific and/or academic documents indexed in Scopus will later be analyzed in a narrative and bibliometric way to deepen the content and possible derivation of common themes that respond directly to the question of research (Rosário and Raimundo, 2021; Raimundo and Rosário, 2021; Rosário et al., 2021).

Of the 60 documents selected, 28 were conference papers, 24 were articles, 4 were reviews, 3 were books, and 1 was a book chapter and short survey.

Publication distribution.

Of peer-reviewed articles on the topic over the period 2014–2021, the maximum in one year was 15, in 2019.

Figure 1 summarizes the published peer-reviewed literature for the 2014–2021 period. The publications were sorted as follows: with four documents, *Computer Law and Security Review* and Proceedings IEEE 2018 International Congress on Cybermatics 2018 IEEE Conferences on Internet of Things Green Computing and Communications Cyber Physical and Social Computing Smart Data Blockchain Computer and Information Technology Ithings GreenCom CPSCom Smartdata Blockchain CIT 2018; with three documents, *International Journal of Recent Technology and Engineering*; with two documents, 2019 IEEE Technology and Engineering Management Conference Temscon 2019; Annual Conference on Innovation and Technology in Computer Science Education ITiCSE; *Journal of Network and Systems Management*; *Journal of Telecommunications and the Digital Economy*; Proceedings of the International Conference on Industrial Engineering and Operations Management); while the other publications had one document each. Interest in the subject has varied over time.

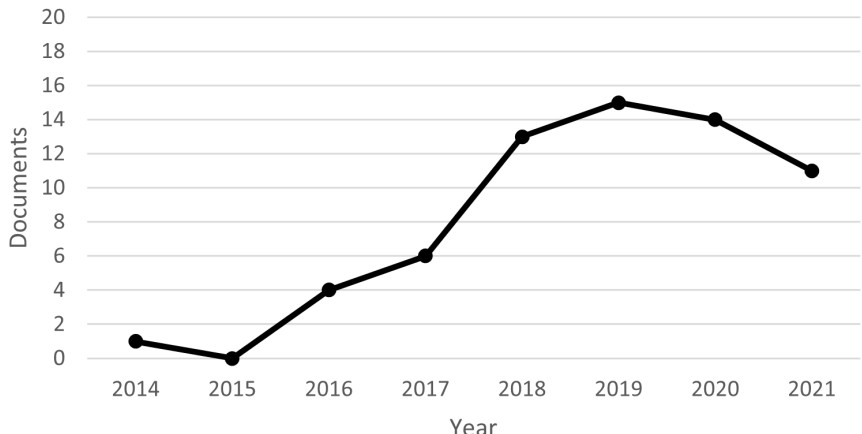

**Figure 1.** The number of documents per year. Source: own elaboration.

In Table 3, we analyze the Scimago Journal and Country Rank (SJR), the best quartile, and the H index by publication.

**Table 3.** Scimago journal and country rank impact factor.

| Title | SJR | Best Quartile | H Index |
|---|---|---|---|
| International Journal of Information Management | 2.770 | Q1 | 114 |
| Journal of Cleaner Production | 1.940 | Q1 | 200 |
| Computer Law and Security Review | 0.820 | Q1 | 38 |
| Technology in Society | 0.820 | Q1 | 51 |
| Business Process Management Journal | 0.670 | Q1 | 81 |
| Advances in Production Engineering and Management | 0.620 | Q1 | 18 |
| ACM Transactions on Management Information Systems | 0.600 | Q1 | 29 |
| Journal of Network and Systems Management | 0.490 | Q2 | 35 |
| International Journal of Automotive Technology and Management | 0.380 | Q2 | 22 |
| Foresight | 0.370 | Q2 | 30 |
| Entrepreneurial Business and Economics Review | 0.330 | Q2 | 11 |
| Vision | 0.310 | Q3 | 9 |
| IEEE Engineering Management Review | 0.300 | Q3 | 20 |
| Managerial Finance | 0.270 | Q3 | 39 |
| International Journal of Business Information Systems | 0.260 | Q3 | 26 |
| Academy of Entrepreneurship Journal | 0.210 | Q3 | 12 |
| Journal of Telecommunications and the Digital Economy | 0.200 | Q2 | 6 |
| Logforum | 0.200 | Q3 | 4 |
| International Journal of Business Analytics | 0.160 | Q4 | 9 |
| International Journal of Computing and Digital Systems | 0.150 | Q4 | 6 |
| International Journal of Technology Intelligence and Planning | 0.130 | Q4 | 15 |
| Economist United Kingdom | 0.100 | Q4 | 9 |
| Petroleum Economist | 0.100 | Q4 | 4 |
| Annual Conference on Innovation and Technology in Computer Science Education ITiCSE | 0.260 | -* | 23 |
| Proceedings 16th IEEE Acis International Conference on Computer and Information Science Icis 2017 | 0.210 | -* | 17 |
| 12th Aeit International Annual Conference Aeit 2020 | 0.190 | -* | 9 |
| Proceedings of the Summer School Francesco Turco | 0.150 | -* | 9 |
| Proceedings of the International Conference on Industrial Engineering and Operations Management | 0.130 | -* | 9 |
| Proceedings of the International Conference on Electronic Business Iceb | 0.120 | -* | 7 |
| 2017 IEEE Technology and Engineering Management Society Conference Temscon 2017 | 0.210 | -* | 6 |
| 2019 IEEE Technology and Engineering Management Conference Temscon 2019 | 0.150 | -* | 4 |
| Ictc 2019 10th International Conference on ICT Convergence ICT Convergence Leading the Autonomous Future | 0.120 | -* | 3 |
| 2018 IEEE Technology and Engineering Management Conference Temscon 2018 | 0.120 | -* | 3 |
| Idimt 2018 Strategic Modeling in Management Economy and Society 26th Interdisciplinary Information Management Talks | 0.100 | -* | 3 |
| International Journal of Recent Technology and Engineering | 0 | -* | 20 |
| Contributions to Management Science | 0 | -* | 14 |

**Table 3.** *Cont.*

| Title | SJR | Best Quartile | H Index |
|---|---|---|---|
| Proceedings IEEE 2018 International Congress on Cybermatics 2018 IEEE Conferences on Internet of Things Green Computing and Communications Cyber Physical and Social Computing Smart Data Blockchain Computer and Information Technology Ithings GreenCom CPSCom Smartdata Blockchain CIT 2018 | -* | -* | -* |
| 2020 IEEE Technology and Engineering Management Conference Temscon 2020 | -* | -* | -* |
| 2020 International Conference on Technology and Entrepreneurship Virtual Icte V 2020 | -* | -* | -* |
| Artificial Intelligence Techniques for a Scalable Energy Transition Advanced Methods Digital Technologies Decision Support Tools and Applications | -* | -* | -* |
| How to Compete in the Age of Artificial Intelligence Implementing a Collaborative Human Machine Strategy for Your Business | -* | -* | -* |
| Innovation Technology in Smart Cities | -* | -* | -* |
| Joint 13th Ctte and 10th Cmi Conference on Internet of Things Business Models Users and Networks | -* | -* | -* |
| Proceedings 18th IEEE International Conference on Machine Learning and Applications Icmla 2019 | -* | -* | -* |
| Proceedings 2019 IEEE 5th International Conference on Collaboration and Internet Computing Cic 2019 | -* | -* | -* |
| Proceedings 2020 IEEE International Conference on Blockchain 2020 | -* | -* | -* |
| Proceedings 2021 21st Acis International Semi Virtual Winter Conference on Software Engineering Artificial Intelligence Networking and Parallel Distributed Computing Snpd Winter 2021 | -* | .* | -* |
| Proceedings of the International Conference on Research Innovation Knowledge Management and Technology Application for Business Sustainability Inbush 2020 | -* | -* | -* |

Note: * data not available. Source: own elaboration.

The *International Journal of Information Management* is the most quoted publication, with 2.770 (SJR), Q1, and H index 114.

There is a total of seven journals in Q1, four journals in Q2, seven journals in Q3, and five journals in Q4. Journals from best quartile Q1 represent 15% of the 48 journals titles, best quartile Q2 represents 8%, best quartile Q3 represents 15%, and best Q4 represents 10%. Data are not available for 25 of the 48 publications, representing 52%.

As is evident from Table 3, the majority of articles on cybersecurity in the Internet of Things in Industrial Management are in the Q1 best quartile index.

The subject areas covered by the 60 scientific articles were: Business, Management, and Accounting (60); Computer Science (31); Engineering (25); Decision Sciences (23); Social Sciences (14); Economics, Econometrics, and Finance (7); Energy (5); Medicine; Environmental Science (3); Mathematics; and Physics and Astronomy.

The most quoted article was "Blockchain technology innovations" with 155 quotes published in the 2017 IEEE Technology and Engineering Management Society Conference, TEMSCON 2017, 0.210 (SJR), not yet assigned quartile, and with H index (6).

The published article focuses on demonstrating the use of Blockchain technology in various industrial applications.

In Figure 2, we can analyze the evolution of citations of articles published between 2014 and 2021. The number of quotes shows a positive net growth with an $R^2$ of 80% for the period 2014–2021, with 2020 reaching 217 citations.

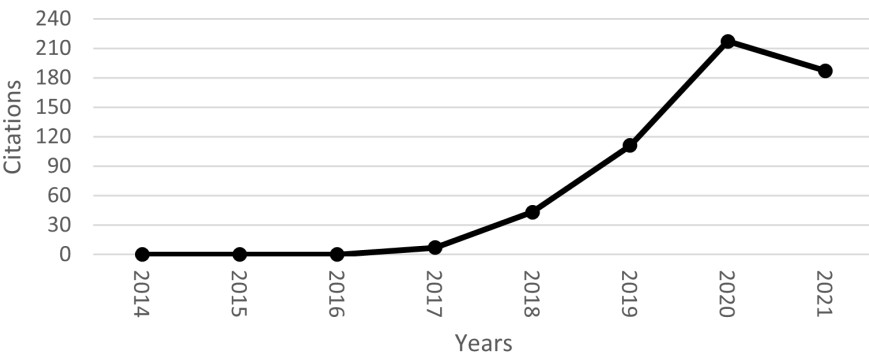

**Figure 2.** Evolution of citations between 2014 and 2021. (Source: own elaboration).

The H index was used to ascertain the productivity and impact of the published work, based on the largest number of articles included that had at least the same number of citations. Of the documents considered for the H index, 10 have been cited at least 10 times.

In Appendix A, the citations of all scientific articles from 2014 to 2021 are analyzed, with a total of 568 citations; of the 60 publications, 19 were not cited. Appendix B examines the self-citation of the document during the 2014 to 2021 period: 20 documents were self-cited 48 times, whereas the article "20 years of scientific evolution of cyber security: A scienc" . . . ", by Furstenau et al. (2020), published in the "Proceedings of the International Conference on Industrial Engineering and Operations Management", was cited 10 times.

In Figure 3, a bibliometric study was performed to examine the development of scientific information by the main keywords. The study of bibliometric outputs by the scientific software VOSviewe aims to identify the main research keywords "Cyber Security" and "Internet of Things".

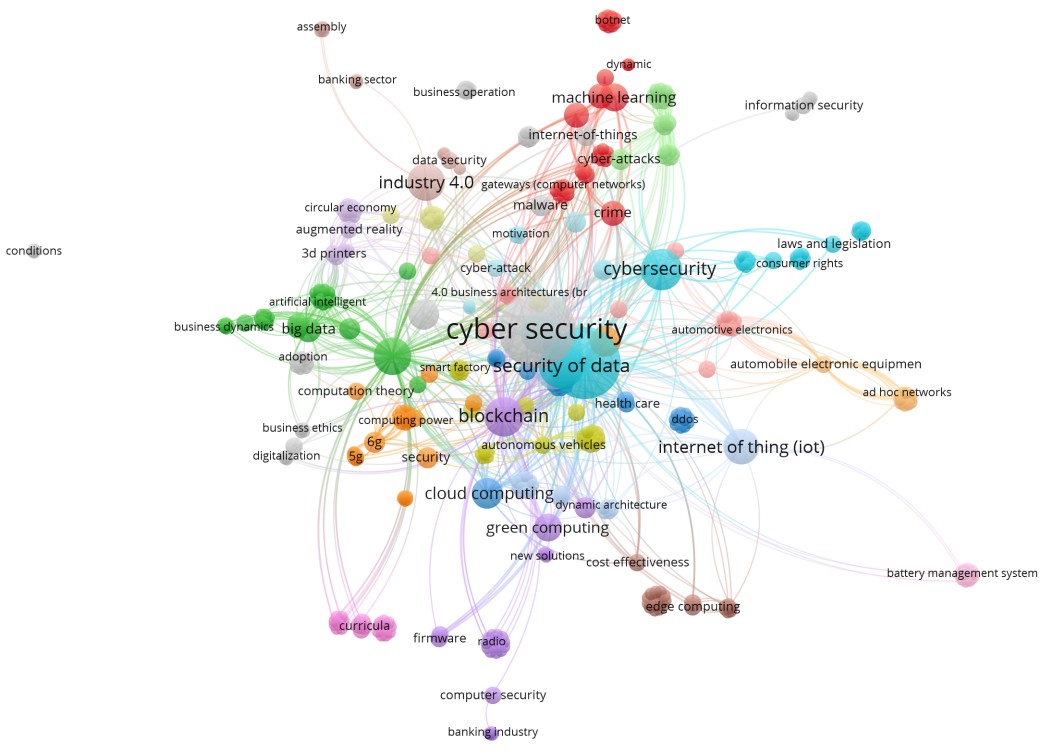

**Figure 3.** Network of all keywords. Source: own elaboration.

The research relied upon the studied articles on consumer marketing strategy on e-commerce in the last decade. The correlated keywords can be viewed in Figure 4, which shows the network of keywords that appear together/linked in each scientific article, as well as knowing the topics studied to identify future research trends. Figure 5 also illustrates co-citations within a unit of analysis of cited references.

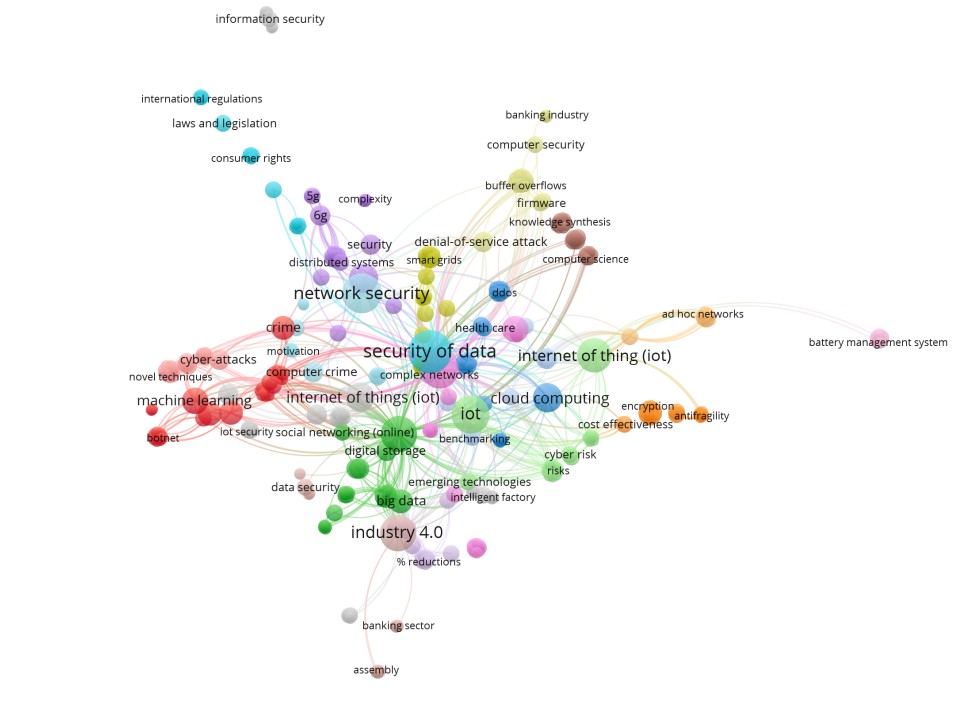

**Figure 4.** Network of linked keywords. Source: own elaboration.

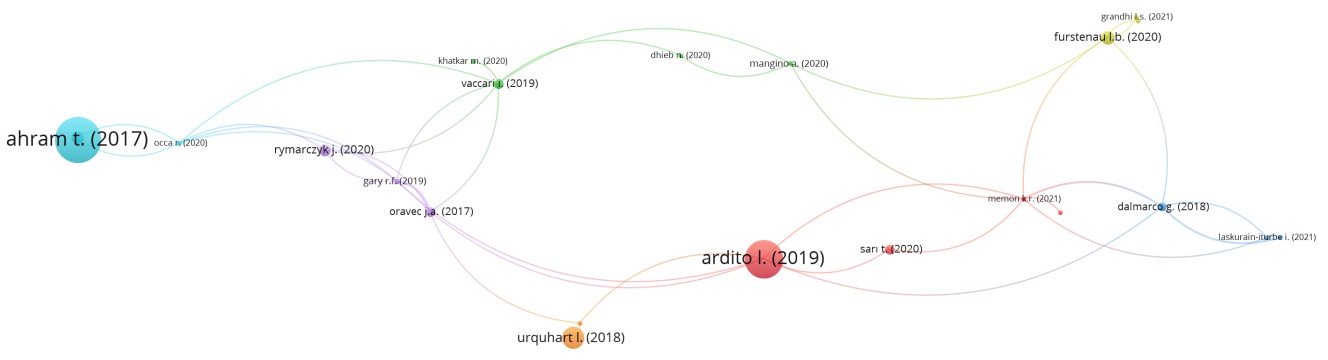

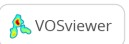

**Figure 5.** Networks of co-citation. Source: own elaboration.

## 4. Discussion

The aforementioned topics in relation to cybersecurity in the IIoT emerge in literature under distinct subthemes, such as machine learning and cloud computing, through several applications related to security. The abovementioned concepts have been widely deployed to solve important issues and highlight principal authors, particularly Ahram [20] and

Ardito [32] (Figure 5). The key themes that underscore the current debate are illustrated in Figures 3 and 4 and need emphasizing.

### 4.1. Cybersecurity

Cybersecurity, as already discussed, has focused primarily on securing distinct data from physical and cloud threats. It deals with cybersecurity threats to digital infrastructure and is a concern for the maintenance of business growth amidst a scenario of changing technologies for social, mobility, analytics, and cloud (SMAC) domains and the Internet of Things (IoT), demanding the validation of new cybersecurity capabilities [39]. It focuses on the users' susceptibility to cyber-attack, the way different factors (e.g., user competence to deal with online threats) mediates the relationship in IoT [40]; it elicits major threat drivers and identifies emerging technologies (e.g., encryption and Blockchain) that are likely to have an impact on defense and attack capabilities in cybersecurity [16].

The existing literature also identifies major platforms that could accommodate smart objects, such as smart home systems that are platforms for connecting sensors, which are consequently exposed to identity theft and need to be protected [1]; the issues of securing automated power consumption units, implemented by smart system technology in an IoT-controlled environment [41]; and reviews the most critical technologies, best practices, policies, and security frameworks in different countries, along with relevant government, industry, civil society and academia [59]. Finally, some areas of the literature examine whether cybersecurity law is justified, analyzing some countries (e.g., China) that need a cybersecurity regime [21].

### 4.2. Machine Learning

Closely related to cybersecurity is the issue of machine learning, which includes artificial intelligence. This theme focuses on intelligence for energy management, including production systems, its cybersecurity in industry 4.0, and the Internet of Things [49]. It is very centered on the interplay between the feature selection and the interpretation steps in a machine learning workflow, aiming at intrusion detection in IoT networks [24]. It often resorts to artificial intelligence (AI) techniques for recognizing a cyber-attack in internet-connected systems domains such as smartphones or robotic factories and for decision-making with the appearance of an incident through data mining approaches, in which AI will improve cyber countermeasures [8]. Others intend to involve AI in business, assisting in adopting a strategy that is rational, relevant, and practical, across enterprise functions including disruptive technologies such as IoT, Blockchain, and cloud computing [36].

### 4.3. The Internet of Things (IoT)

The IoT is central to this debate and its influence extends to industry (IIoT) and network and cloud computing. This issue has shed light on the implementation of intrusion detection systems that are able to protect data and physical devices. They do so through AI that allows an intelligent intrusion detection model to expose threats, through Decision Trees for network intrusion detection [23]. It also enables household participants to obtain control over intelligent IoT agents operating in their personal spheres [60]; executives to provide businesses with an approach for securing an enterprise by a dynamic architecture of Extended Risk-Based Approach on Cloud and the IoT [61]; and in general, to virtually all paperless work environments [62], by targeting threats related to the distributed denial of service (DDoS) on power grids and hacking of industrial control systems (ICS) along with the ensuing regulatory responses [13].

IoT theory also searches for solutions related to how supply chains as a whole may benefit from the adoption of 4.0 technologies, delivering the flexible response customers want, and benefiting from big data, cloud computing, and cybersecurity by an improved communication system [30]. For example, it aims to analyze devices and network security,

while considering different scenarios involving various attackers aiming to destroy the IoT wireless network [17], plus applying learning curves to major global cyber-attacks [42].

A different stream of literature explores which technologies are being deployed and where the organizational risk is being considered within the organization, building a risk model to deal with AI, IoT, and distributed ledger [43], while offering a detailed study of trust management models to enforce different security measures in IoT system and ensuring safety to connected devices [44]. This includes technologies such as augmented reality (AR), a concept that connects the real world to the virtual world, to develop guidelines for industry 4.0. [31], tourism, in integrating business, and key performance metrics [63].

The IoT thus embraces a myriad of smart objects connected to the network; it needs to be both safe, with regard to physical objects and people, and secure with respect to data and IT systems. In so doing, there is an attempt to secure the automotive industry by developing an encryption system that is able to disguise data to those other than the user and detect any set of actions that aim to compromise the vehicle. In particular, an intrusion detection system capable of analyzing traffic over the Controller Area Network (CAN-Bus) and of understanding whether the messages that transmit over the communication channel are malicious or not [64–66].

It is worth mentioning the case of electric vehicles' (EVs) cybersecurity issues in identifying the key matters that have been developed, but do not address the requirements of cybersecurity (e.g., EV battery stacks) [3] on networks [4] and suggest strategies the automotive industry might pursue in this regard to face cybersecurity threats [28]. In the same vein, others detail the security vulnerabilities of unmanned ships, subsequent defense strategies, and ensuing countermeasures [55], while signaling vulnerabilities of wireless systems of software radios [45].

To summarize, as smart devices are growing in number, there is a corresponding growth in risks, both to the user and to the Internet as a whole from hacking threats [54]. However, there is a lack of policy direction, user values on cybersecurity are misunderstood, and there is a lack of clarity as to how IoT public policy should be developed in terms of, for instance, being guided by stakeholder values [10]. Moreover, a new paradigm of proactive antifragility for cyber defense approaches is demanded in the IoT, namely in distributed highly complex computing paradigms beyond traditional cyber defense (e.g., the Internet of Battle Things (IoBT) [9]), in order to cope with novel threats [53].

Finally, others propose solutions to new wireless challenges such as 6G, dynamic spectrum access (DSA) [67], and wireless mesh networks (WMNs) [68]. It allows shifting the paradigm "from the Internet of Things (IoT) to Internet of Intelligence (IoI)" to provide connectivity, while maintaining the ability to process knowledge and make decisions autonomously [47]. It aims to develop a novel methodology for fingerprinting IoT devices by building data-driven techniques rooted in machine learning methods, which allows unveiling compromised IP addresses throughout diverse geographical areas [5]. Moreover, and since there is a lack of effective IoT cyber risk management frameworks [69], of effective sensor networks [70], it raises the issue of a distributed denial of service (DDoS) in terms of classifications and opportunities for attacks, particularly in the health sector, in areas of limited security [22], while examining the changing legal environment in a IoT regulatory context [48].

### 4.4. Industry 4.0 (IIoT)

Industry 4.0 is another important subtheme related to the IoT. It is also known as the Industrial Internet of Things (IIoT). Part of the literature investigates the influence of critical technologies such as artificial intelligence, big data, and virtual augmented reality on the circular economy (e.g., recycling and reduction of waste and emissions), which confirms the importance of Industry 4.0 for improving circularity [18]. Others enhance the impact of those digital technologies on e-finance, an opportunity to change business models, for example through AI [27], while developing these technologies for managing the interface between supply chain management and marketing processes in sustaining

supply chain management marketing (SCM-M) integration [32]. Lastly, it focuses on the oil and gas industry in terms of threats to the migration of sensitive business data to cloud digital platforms in industrial processes, which include decision-making processes and procedures [33] and the increase in entry points for organizations to defend themselves against threats [34].

Another stream of literature spots the research gaps in industry 4.0, using an open (Google) internet-based research search engine (OIBRSE) to acquire the digital object identifiers and universal resource locators if the DOI does not exist with research articles [53]. There are current debates around this, for example, the issue of smart factory, which is based on ICT technology and used to drive down manufacturing costs and time, while security vulnerabilities must be reduced [6]. Furthermore, with regard to this sort of critical infrastructure, securing electro-energy platforms represents an important demand for a secure electric vehicle recharge system monitoring and control platform [52].

The main issue is always centered on the way cybersecurity deals with cyber-attacks for industry 4.0, while mapping current topics such as "cloud computing", "smart grid", "intrusion detection", "privacy", "Internet of Things", and "smart cities" [11]. It is necessary to keep up with the technological paradigm shift, while introducing measures that prevent significant expected fatalities [51], over disparate sectors from e-commerce to banks. Also, emphasizing how to cope with digitalization, while making customers a priority [37] among different manufacturers [7]. In the end, the Industrial Revolution 4.0 will have economic, social, and political consequences at global level, leading to revolutionary changes in the intelligent processes of goods production and services, with associated rising unemployment and social stratification [38].

### 4.5. Blockchain and Cloud Computing

The decentralized architectures of IoT devices are a current debate. The latest developments in Blockchain technologies, for example, have allowed the use of a smart ecosystem able to support cybersecurity mechanisms across distinct sectors such as smart home installations, focusing on the immutability of users and devices as well as the dynamic and immutable management of blocked malicious IPs [25]; in multiple industrial applications, healthcare, finance, and government [20] in terms of solutions for cybersecurity problems such as accountability, traceability, and identification [12].

The key point is how to improve the security of systems architecture to protect against malicious internal users and malware implanted inside the system, which can be solved through an effective sensor network [70], inherent to Blockchain security architecture [19]. Blockchain may contribute to privacy, security, and non-repudiation of an IoT system, through the large amount of data generated and variety of sensors and devices adopted [2], as Blockchain technology builds a scalable and decentralized end-to-end secure IoT system [14]. Furthermore, the IoT can be enhanced with an AI at the gateway level to detect and classify suspected activities [14]. Moreover, Blockchain technology is also of use in parallel with cloud computing for higher education, in terms of primary infrastructure topology, putting together machine learning and artificial intelligence on training opportunities [35].

Cloud computing is therefore highly correlated with Blockchain technology, namely in preventing attacks against, for example, radio-frequency (RF)-enabled hardware, Internet of Things (IoT) firmware, and wireless protocols [50]. Interconnectedness of intelligent devices and the use of public networks is at the center of debate in smart cities due to interconnected services for their citizens, in which cybersecurity has become a major concern [29], namely in issues such as communication infrastructures, cloud computing, smart health, and energy management [26].

The discussion on cloud computing covers subthemes of cybersecurity of supply chains based on software and networks, to minimize risks of purchasing and disconnection of key machines from networks [46]. To summarize, 5G and 6G networks can provide novel communication networks infrastructure, although IoT systems will still have the same energy capacity for hackers to take advantage of these weaknesses. There is a need

for a system that may identify and counter potential threats in next generation networks and decentralized systems such as Blockchain [15].

## 5. Conclusions

The IoT has been a key element for smart manufacturing, smart cities, smart health, smart grids, and EVs, for example. IoT and IIoT bridges physical artifacts and the Internet, both in our daily lives and in the industry environment. On the one hand, such linkage unveils countless opportunities, while on the other hand it exposes our information and behaviors to potentially hacking sensitive data and critical infrastructure.

Additionally, the IoT produces huge amounts of information that need to be protected and that is linked to varied security risks regarding its interconnectedness, either through cloud computing or Blockchain in smart factories, smart homes, and smart cities, for instance. Thus, due to the need of decision-making and investment, cybersecurity must first focus on the varying weaknesses of IoT objects and further work on its security mechanisms such as privacy, access control, data storage, and authorization, whereas organizations should adopt a cybersecurity strategy. Organizations need to keep up with the development of technologies to respond appropriately to cybersecurity threats. This study fills a gap in IIoT cybersecurity literature because it comprehensively explains its main subthemes, while encouraging further research on the topic.

Furthermore, arising technologies such as Blockchain may perform a central role in the future of cybersecurity in IoT and the IIoT, whilst security will become more important in the future because of the increase in the number of cordless-connection objects in the short term, which extends to virtually all areas of our daily lives that need to be effectively managed.

**Author Contributions:** Conceptualization, R.J.R.; A.T.R.; methodology, R.J.R.; A.T.R.; software, R.J.R.; A.T.R.; validation, R.J.R.; A.T.R.; formal analysis, R.J.R.; A.T.R.; investigation, R.J.R.; A.T.R.; resources, R.J.R.; A.T.R.; data curation, R.J.R.; A.T.R.; writing—original draft preparation, R.J.R.; A.T.R.; writing—review and editing, R.J.R.; A.T.R.; visualization, R.J.R.; A.T.R.; supervision, R.J.R.; A.T.R.; project administration, R.J.R.; A.T.R.; funding acquisition, R.J.R.; A.T.R. All authors have read and agreed to the published version of the manuscript.

**Acknowledgments:** We would like to express our gratitude to the editor and the referees. They offered extremely valuable suggestions and improvements. The authors were supported by the GOYCOPP Research Unit of Universidade de Aveiro and ISEC Lisboa, Higher Institute of Education and Sciences.

**Funding:** This research received no external funding.

**Conflicts of Interest:** The authors declare no conflict of interest.

## Appendix A

**Table A1.** Overview of document citations from 2014 to 2021.

| Documents | | 2014 | 2015 | 2016 | 2017 | 2018 | 2019 | 2020 | 2021 | Total |
|---|---|---|---|---|---|---|---|---|---|---|
| An Intelligent Tree-Based Intrusion Detection Model for Cybe... | 2021 | - | - | - | - | - | - | - | 1 | 1 |
| User values and the development of a cybersecurity public po... | 2021 | - | - | - | - | - | - | 1 | 1 | 2 |
| A Security-UTAUT Framework for Evaluating Key Security Deter... | 2021 | - | - | - | - | - | - | - | 1 | 1 |
| On Australia's cyber and critical technology international e... | 2020 | - | - | - | - | - | - | - | 1 | 1 |
| Internet-scale Insecurity of Consumer Internet of Things | 2020 | - | - | - | - | - | - | 1 | 1 | 2 |
| A Blockchain Solution for Enhancing Cybersecurity Defence of... | 2020 | - | - | - | - | - | - | - | 1 | 1 |
| Scalable and Secure Architecture for Distributed IoT Systems | 2020 | - | - | - | - | - | - | - | 1 | 1 |
| Modeling for malicious traffic detection in 6G next generati... | 2020 | - | - | - | - | - | - | 3 | 1 | 4 |
| Awareness and readiness of Industry 4.0: The case of Turkish... | 2020 | - | - | - | - | - | - | 1 | 6 | 7 |
| Technologies, opportunities and challenges of the industrial... | 2020 | - | - | - | - | - | - | 2 | 8 | 10 |
| An overview of distributed denial of service and internet of... | 2020 | - | - | - | - | - | - | - | 1 | 1 |
| Artificial intelligence techniques for a scalable energy tra... | 2020 | - | - | - | - | - | - | - | 1 | 1 |
| 20 years of scientific evolution of cyber security: A scienc... | 2020 | - | - | - | - | - | - | 12 | 3 | 15 |
| A review on the research growth of industry 4.0: IIoT busine... | 2020 | - | - | - | - | - | - | - | 4 | 4 |
| Toward a cloud computing learning community | 2019 | - | - | - | - | - | - | 1 | 2 | 3 |
| Towards the integration of a post-hoc interpretation step in... | 2019 | - | - | - | - | - | - | - | 2 | 2 |
| Proactive antifragility: A new paradigm for next-generation... | 2019 | - | - | - | - | - | - | - | 1 | 1 |
| A Research on the Vulnerabilities of PLC using Search Engine | 2019 | - | - | - | - | - | - | 1 | - | 1 |
| Cyber security threat intelligence using data mining techniq... | 2019 | - | - | - | - | - | - | 2 | 1 | 3 |
| Addressing Industry 4.0 Cybersecurity Challenges | 2019 | - | - | - | - | - | 1 | 9 | 12 | 22 |
| FACTS approach to address cybersecurity issues in electric v... | 2019 | - | - | - | - | - | 4 | 3 | 3 | 10 |
| Towards Industry 4.0: Mapping digital technologies for suppl... | 2019 | - | - | - | - | - | 12 | 50 | 47 | 111 |
| Evaluating security of low-power internet of things networks | 2019 | - | - | - | - | - | 1 | 7 | - | 8 |
| Legitimate firms or hackers—who is winning the global cybe... | 2019 | - | - | - | - | - | - | 2 | 1 | 3 |
| Foresight of cyber security threat drivers and affecting tec... | 2018 | - | - | - | - | - | 1 | 3 | 5 | 9 |
| Agile Business Growth and Cyber Risk: | 2018 | - | - | - | - | - | 1 | 1 | - | 2 |

**Table A1.** *Cont.*

| Documents | | 2014 | 2015 | 2016 | 2017 | 2018 | 2019 | 2020 | 2021 | Total |
|---|---|---|---|---|---|---|---|---|---|---|
| How to compete in the age of artificial intelligence: Implem... | 2018 | - | - | - | - | - | 1 | 1 | 2 | 4 |
| Solving Global Cybersecurity Problems by Connecting Trust Us... | 2018 | - | - | - | - | - | 1 | - | 1 | 2 |
| A Cybersecurity Case for the Adoption of Blockchain in the F... | 2018 | - | - | - | - | - | - | 1 | 1 | 2 |
| Cybersecurity Attacks and Defences for Unmanned Smart Ships | 2018 | - | - | - | - | - | - | 5 | 1 | 6 |
| Avoiding the internet of insecure industrial things | 2018 | - | - | - | - | 5 | 14 | 8 | 9 | 36 |
| The impact of China's 2016 Cyber Security Law on foreign tec... | 2018 | - | - | - | - | 6 | 3 | 6 | 5 | 20 |
| Adoption of industry 4.0 technologies in supply chains | 2018 | - | - | - | - | - | 1 | 3 | 2 | 6 |
| Artificial intelligence in smart tourism: A conceptual frame... | 2018 | - | - | - | - | - | 1 | 2 | 5 | 8 |
| Cybersecurity and the auto industry: The growing challenges... | 2018 | - | - | - | - | - | 3 | 5 | 1 | 9 |
| Information innovation technology in smart cities | 2017 | - | - | - | - | - | 3 | - | - | 3 |
| Kill switches, remote deletion, and intelligent agents: | 2017 | - | - | - | - | 2 | - | 2 | 3 | 7 |
| Blockchain technology innovations | 2017 | - | - | - | - | 11 | 46 | 60 | 37 | 155 |
| Personality traits and cyber-attack victimisation: Multiple... | 2017 | - | - | - | - | - | - | 4 | - | 4 |
| STM32-based vehicle data acquisition system for Internet-of-... | 2017 | - | - | - | 1 | 2 | 4 | 5 | 5 | 17 |
| Electronic finance—recent developments | 2017 | - | - | - | 2 | 1 | - | 6 | 3 | 12 |
| Cybersecurity in the Internet of Things: Legal aspects | 2016 | - | - | - | 4 | 16 | 14 | 10 | 7 | 51 |
| | Total | 0 | 0 | 0 | 7 | 43 | 111 | 217 | 187 | 568 |

## Appendix B

**Table A2.** Overview of document self-citations from 2014 to 2021.

| Documents | | 2014 | 2015 | 2016 | 2017 | 2018 | 2019 | 2020 | 2021 | Total |
|---|---|---|---|---|---|---|---|---|---|---|
| A Security-UTAUT Framework for Evaluating Key Security Deter... | 2021 | - | - | - | - | - | - | - | 1 | 1 |
| On Australia's cyber and critical technology international e... | 2020 | - | - | - | - | - | - | - | 1 | 1 |
| Internet-scale Insecurity of Consumer Internet of Things | 2020 | - | - | - | - | - | - | 1 | - | 1 |
| Modeling for malicious traffic detection in 6G next generati... | 2020 | - | - | - | - | - | - | 2 | - | 2 |
| Technologies, opportunities and challenges of the industrial... | 2020 | - | - | - | - | - | - | - | 1 | 1 |
| 20 years of scientific evolution of cyber security: A scienc... | 2020 | - | - | - | - | - | - | 10 | - | 10 |
| Toward a cloud computing learning community | 2019 | - | - | - | - | - | - | - | 1 | 1 |
| Towards the integration of a post-hoc interpretation step in... | 2019 | - | - | - | - | - | - | - | 1 | 1 |
| Addressing Industry 4.0 Cybersecurity Challenges | 2019 | - | - | - | - | - | - | - | 1 | 1 |
| Emerging technologies and risk: How do we optimize enterpris... | 2019 | - | - | - | - | - | 4 | 3 | 1 | 8 |
| FACTS approach to address cybersecurity issues in electric v... | 2019 | - | - | - | - | - | 1 | - | 1 | 2 |
| Towards Industry 4.0: Mapping digital technologies for suppl... | 2019 | - | - | - | - | - | - | 5 | - | 5 |
| Legitimate firms or hackers—who is winning the global cybe... | 2019 | - | - | - | - | - | - | 1 | 1 | 2 |
| Solving Global Cybersecurity Problems by Connecting Trust Us... | 2018 | - | - | - | - | - | - | - | 1 | 1 |
| Avoiding the internet of insecure industrial things | 2018 | - | - | - | - | - | 2 | - | - | 2 |
| Adoption of industry 4.0 technologies in supply chains | 2018 | - | - | - | - | - | 1 | - | - | 1 |
| Cybersecurity and the auto industry: The growing challenges... | 2018 | - | - | - | - | - | - | 1 | - | 1 |
| Blockchain technology innovations | 2017 | - | - | - | - | 1 | 1 | - | 1 | 3 |
| Electronic finance—recent developments | 2017 | - | - | - | - | - | - | 2 | - | 2 |
| Cybersecurity in the Internet of Things: Legal aspects | 2016 | - | - | - | 1 | - | 1 | - | - | 2 |
| Total | | - | - | - | 1 | 1 | 10 | 25 | 11 | 48 |

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
