# Peer review of "Cybersecurity in the Internet of Things in Industrial Management"

_applsci, doi:10.3390/app12031598_

Round 1

Reviewer 1 Report

1.Author did not mention their exact contribution.

2. lack of literature review, consider the recently work and analyzed the limitation existing work.

3.Cybersecurity in the Internet of Things , This is a very strong claim and is not reflected in the paper. In my humble opinion, this analysis is just a generalization, without delving into the details of what is required in a real-world deployment or how this can be integrated into a large-scale system. F

4. figure 3 to 5 need more explanation

Author Response

Dear reviewer 1,

  • The aim of this article is to present the current debate around the issue of IIoT, rather than suggesting any particular technical solutions to solve network security problems, as stated now in the abstract.
  • Literature review was based on a review of 65 key articles that came up from a profound Scopus literature survey.
  • IoT was extensively debated at section 4.3, throughout the references that came across the Scopus lit review.
  • The discussion section dealt with the major clusters arisen from the Scopus Lit review and Lexmanser graphics at fig. 3 to 5

Reviewer 2 Report

Comments inserted via an uploaded word document

Author Response

Thanks

Reviewer 3 Report

The research concerns an important topic. It is performed with effective scientific tools for constructing a review. This approach makes it possible to cover and explore the broad topic of the work. This is the reason it is thorough and well-founded.

Author Response

Dear Reviewer 2,

Thanks for the feedback. We addressed the suggestions proposed the best we could. In so,

  • The style and flow in terms of grammar and articulation was ameliorated;
  • The research goals were enhanced, in particular the attempted literature review, based on a Scopus search, regarding diverse streams of literature and without proposing any technical solution in particular to solve network security problems.
  • The suggested references were added to the initial review.

Many thanks,

RR

Reviewer 4 Report

Overall the paper seems to be interesting and sound, however, in my opinion, it is still affected by some problems.

First of all, in some parts, the clarity and editorial quality of the paper weaken. As a consequence, such parts result to be quite difficult to read. Therefore, I would suggest to carefully improve the prose of writing in order to make this paper easier to read.

Furthermore, presentation aside, by reading the paper, it still was not entirely clear what to expect with the direction of the article. Indeed, the contribution proposed in this paper has been only marginally compared and contextualised with respect to the state of the art. As a result, it is extremely difficult to understand the novelty/contributions introduced by the paper. The aforementioned aspects should be carefully addressed before the paper can be considered any further.

The paper should be better compared and contextualized with respect to the state of the art. I want suggest these papers to authors:

- Pascale, F.; Adinolfi, E.A.; Coppola, S.; Santonicola, E. Cybersecurity in Automotive: An Intrusion Detection System in Connected Vehicles. Electronics 202110, 1765. 

- Lombardi, M., Pascale, F., and Santaniello, D. (October 26, 2021). "Two-Step Algorithm to Detect Cyber-Attack Over the Can-Bus: A Preliminary Case Study in Connected Vehicles." ASME. ASME J. Risk Uncertainty Part B.

The figures should be better explained in their component parts

Finally, a thorough proofreading would be suggested, since in the paper there are some typos and formatting issues.

As remarks:

  • The paper should be better compared and contextualized with respect to the state of the art.
  • In some parts of the paper, the clarity and editorial quality of the paper weaken. As a consequence, such parts result to be quite difficult to read. Therefore, I would suggest to carefully improve the prose of writing in order to make this paper easier to read.
  • Each figure should be properly defined within the text and must be improved in quality.
  • An accurate proofreading is strongly recommended.

Author Response

Dear reviewer 4,

  • The text was improved;
  • A proofreading was performed.
  • The aim of this article is to present the current debate around the issue of IIoT, rather than suggesting any particular technical solutions to solve network security problems, as stated now in the abstract.
  • It was added the contributions of Pascale, F.; Adinolfi, E.A.; Coppola, S.; Santonicola, E. Cybersecurity in Automotive: AnIntrusion Detection System in Connected Vehicles. Electronics 2021, 10, 1765; and Lombardi, M., Pascale, F., and Santaniello, D. (October 26, 2021). "Two-Step Algorithm toDetect Cyber-Attack Over the Can-Bus: A Preliminary Case Study in Connected Vehicles."ASME. ASME J. Risk Uncertainty Part B, with corresponding references [64] and [65].
  • The thematic clusters in figures are extensively explained, in the light of Scopus literature review

Reviewer 5 Report

This review article mainly discusses the opportunities and threats in IIoT network security. There are no technical solutions to these security problems, more like a popular science article. There are following main problems:
(1) In the Abstract of the paper, a lot of space is about the security problems of the Internet of Things system in general terms, but it is impossible for the reviewers to know what specific research work the authors have done.
(2) In Section 2. Literature Review: key concepts, the content of the article is the introduction of some common terms, more like popular science articles.
(3) In Section 3 "2. Materials and Methods", the numbering is wrong, and the research methods of this paper are described. There is not much technical content.
(4) In "4. Discussion",  no specific technical solutions are put forward to solve the network security problem.
(5) There are many language errors in this paper, such as:
In Table 1, The word "Communicatio" is wrong.
In row 377, 389, 440, et al., the end ";" should be ".".
In row 490,  the end "}" should be "].".
In row 491, The extra space at the beginning should be deleted.
In a word, the full-text content is lacking in technical content, which is not suitable for publication in this scientific journal.

Author Response

Dear reviewer 5,

  • The aim of this article is to present the current debate around the issue of IIoT, rather than suggesting any particular technical solutions to solve network security problems, as stated now in the abstract, based on a review of 65 key articles that came up from a profound Scopus literature survey.
  • Numbering was corrected.
  • The text was improved;
  • A proofreading was performed.

Round 2

Reviewer 1 Report

check the typos errors in whole manuscript

Author Response

Thanks

Reviewer 4 Report

The suggested corrections have been made. I advise authors to read the paper to correct some typos

Author Response

Thanks

Reviewer 5 Report

The author's revised draft has carefully revised and sincerely answered my previous concerns, and many previous mistakes have been corrected. Especially, a large number of references have been revised. Considering the characteristics of the review articles, the paper has a large amount of literature research work, which is of good reference value to the peers in the relevant neighborhood. I changed my decision and suggest Accept this paper after minor revision. A little minor repair suggestion: there are many blank areas above and below many pictures that should be deleted.

Author Response

Thanks